# Slippery Liquid-Infused Porous Polymeric Surfaces Based on Natural Oil with Antimicrobial Effect

**DOI:** 10.3390/polym13020206

**Published:** 2021-01-08

**Authors:** Salma Habib, Sifani Zavahir, Aya E. Abusrafa, Asma Abdulkareem, Patrik Sobolčiak, Marian Lehocky, Daniela Vesela, Petr Humpolíček, Anton Popelka

**Affiliations:** 1Center for Advanced Materials, Qatar University, Doha P.O. Box 2713, Qatar; salma.m.habib@hotmail.com (S.H.); fathima.z@qu.edu.qa (S.Z.); aya_abusrafa94@yahoo.com (A.E.A.); asma.alkareem@qu.edu.qa (A.A.); patrik@qu.edu.qa (P.S.); 2Centre of Polymer Systems, Tomas Bata University in Zlin, Trida Tomase Bati 5678, 760 01 Zlin, Czech Republic; lehocky@post.cz (M.L.); dvesela@utb.cz (D.V.); humpolicek@utb.cz (P.H.); 3Faculty of Technology, Tomas Bata University in Zlin, Vavreckova 275, 760 01 Zlin, Czech Republic

**Keywords:** slippery surface, electrospinning, oil infusion, plasma treatment, antimicrobial activity

## Abstract

Many polymer materials have found a wide variety of applications in biomedical industries due to their excellent mechanical properties. However, the infections associated with the biofilm formation represent serious problems resulting from the initial bacterial attachment on the polymeric surface. The development of novel slippery liquid-infused porous surfaces (SLIPSs) represents promising method for the biofilm formation prevention. These surfaces are characterized by specific microstructural roughness able to hold lubricants inside. The lubricants create a slippery layer for the repellence of various liquids, such as water and blood. In this study, effective antimicrobial modifications of polyethylene (PE) and polyurethane (PU), as commonly used medical polymers, were investigated. For this purpose, low-temperature plasma treatment was used initially for activation of the polymeric surface, thereby enhancing surface and adhesion properties. Subsequently, preparation of porous microstructures was achieved by electrospinning technique using polydimethylsiloxane (PDMS) in combination with polyamide (PA). Finally, natural black seed oil (BSO) infiltrated the produced fiber mats acting as a lubricating layer. The optimized fiber mats’ production was achieved using PDMS/PA mixture at ratio 1:1:20 (g/g/mL) using isopropyl alcohol as solvent. The surface properties of produced slippery surfaces were analyzed by various microscopic and optics techniques to obtain information about wettability, sliding behavior and surface morphology/topography. The modified PE and PU substrates demonstrated slippery behavior of an impinged water droplet at a small tilting angle. Moreover, the antimicrobial effects of the produced SLIPs using black seed oil were proven against Gram-positive *Staphylococcus aureus* (*S. aureus*) and Gram-negative *Escherichia coli* (*E. coli*).

## 1. Introduction

Many biomedical polymers have been found in bioscience paralleling advancements in the technology sector [1,2]. Selection of suitable polymers for use in the biomedical sector is based on many factors, such as chemical nature, surface free energy and morphology, which influence cell–polymer surface interactions [3]. However, these materials carrying infectious agents represent a serious issue for their applications [4]. The resulting infections closely relate with biofilm formation, whereby organisms are strongly attached to a surface forming strongly attached multicellular communities [5]. These biofilms formed on the various surfaces have remarkable resistance to their removal because of their sheer resistance [6]. Microbial biofilm formation is responsible for remarkable problems predominantly in clinical adjusting. Biofilms lead to over 80% of infections caused by microbes in the human body [7]. Pathogenic types of biofilm are resistive against destruction by the immune system/antibiotics, leading to serious infections [8,9,10]. Moreover, bacterial pathogens have become insensitive to the antibiotic therapies, making biofilms more resistant [11]. Therefore, a non-cytotoxic treatment is based on a physical approach—preventing the bacterial adhesion; this can avoid the problems associated with drug resistance [12,13]. Infections are closely related to biofilm formation, and the reduction of bacterial colonization has great importance for medical materials in healthcare systems. One example is urinary tract infections being about 40% of all nosocomial infections [14]. The detailed mechanism of the biofilm formation can be divided into five steps. In the first step, the initial stage is responsible for reversible interactions of the bacteria and surface. However, after the bacterial detachment, some of the extracellular traces from substances can remain on the surface and they are predominately responsible for the biofilm formation. The second step represents irreversible attachment of bacteria and leads to stable bacterial attachment on the substrate surface (primary adhesion effect). In the subsequently phase, micro-colonies are formed by reproduction processes of the attached microorganisms. In the next step, a biofilm is formed when the adhered microorganisms produce an extracellular secretion, and eventually the substrate surface can be fully covered by this secretion. This biofilm is suitable for the reproduction of microorganisms. The final step is represented by the distribution of the biofilm as a result of the growing mass, and present bacteria can be spread into the surrounding environment [15]. Several modifications are known for the prevention of biofilm formation on medical polymeric materials, such as surface modification, antibacterial agent mixing and copolymerization [16,17]. The most suitable means of biofilm prevention is only surface modifications because only top layer is affected, without any negative effects on bulk properties [18,19]. Various strategies can be used for surface modification—for example, repellent coating prepared by chemical modification, slowly releasing antibiotics from the surface and using quaternary ammonium compounds or noble metals (silver) as antimicrobial agents [20,21,22]. However, the problem is with nano-toxicity and many studies have confirmed destructive effects on cells at higher concentrations of silver. The repellent coating represents the most pronounced modification method for the biofilm prevention. Polymer brushed structures are often used as effective microbial-repellent coatings, such as linear, branched and star-shaped ones [23,24]. Coatings consisting of high-density brushes have significant anti-adhesive properties and the chains length is a very important factor for the biofilm prevention; longer chains are usually more effective than shorter ones [25,26]. Many studies have investigated the effects of hydrophobicity and hydrophilicity on bacterial adhesion [27,28,29]. A hydrophobic surface is more susceptible to the adherence in the initial stage, but the subsequently removal of microorganisms is easier compared with hydrophilic surfaces. Surface topography closely relates to hydrophobic and hydrophilic properties, and therefore it is also significant for the prevention of microorganism adhesion [30,31]. The mucus layer covering the intestinal tract responsible for the shielding and protection of tissues from bacterial adhesion was the inspiration for anti-biofouling surfaces. A liquid layer deposited on the surface with high roughness is responsible for the low adhesion. These slippery liquid-infused porous surfaces (SLIPSs) represent promising tools for antibacterial modifications of different types of materials [32,33,34,35,36,37]. Polyurethane (PU) and polyethylene (PE) are the most suitable substrates used in medical applications, such as implantable devices and tubing, but with some limitations [38,39]. For example, PU is biodegradable, and this affects its stability after implantation [40]. Moreover, its surface is vulnerable towards microbial attacks and this allows accumulation of bacterial colonies on it and may cause some severe inflammation in the implanted area. Interestingly, polydimethylsiloxane (PDMS) is found to be a good coating candidate to prevent the biodegradation due to its inert properties and biocompatibility. PDMS is well known for its excellent surface properties. It is an affordable polymer with biostable, biocompatible, and permeable characteristics and good hydrophobicity. It is usually used in coatings and dressings [41]. However, due to its relatively low molecular weight and the method applied for crosslinking, it has only a weak ability to form proper electrospun fibers. However, it can be incorporated effectively to another polymer that can form electrospun fibers as a polymer mixture. Correspondingly, polyamide (PA) is well-known known polymer having a great applicability for the production of strong fibers that are used in various applications as the water filtration, wound dressing, etc. [42,43,44]. PA is characterized as biocompatible, and resistant to humidity and water [45]. PDMS can be mixed with the previous mentioned polymers to prepare electrospun fibers with synergic final properties. An effective procedure can be applied for the fiber preparation by electrospinning using a combination of two polymers—the mixing of polymer solutions before electrospinning and using it as a blend. The polymer mixture/blend is prepared by dissolving the polymers in one solvent that is able to dissolve both or by using two miscible solvents and then electrospinning them as one blend. A further application is to apply electrospun fibers on a polymer substrate as the collector for the formation of a coating. By the preparation of the electrospun fibers made of PDMS on the PU surface, they can act as a coating, enhancing the PU used in medical applications and improving the biodegradation inhibition [46]. Several modifications are known for the prevention of biofilm formation on different medical polymeric materials, such as surface modifications, antibacterial agent mixing and copolymerization [16,17]. The most suitable way for the biofilm prevention is surface modifications because only the top layer is affected—hence, there are no negative effect on bulk properties [18,19]. In healthcare devices, such as catheters, biofouling control is highly regarded, as pathogens otherwise accumulate and spread diseases. One easy way to overcome this issue is by keeping the surfaces dry and sliding away any liquid drops incident on the surfaces. This approach is more environmentally friendly than biocidal coatings. SLIPs [33,47] is the most recent improvement, addressing weaknesses caused by super hydrophobic surfaces [48,49]. Super hydrophobic surfaces have micro/nano structures that are capable of holding an air layer in between and preventing the surface from wetting by the liquid. For SLIPSs, a lubricating thin film coated on the surface makes sure the liquid droplets on the surface slide away. This concept was inspired by the natural pitcher plant [50], which has a thin lubricating layer on the microscopic rough structures on the peristome. This lubricating film reduces the friction, so any small organism incident on the petal slides to the digestive track of the plant.

In this study, to achieve bacteria-free surfaces by inhibiting contaminant attachment with antimicrobial activity [51,52,53], SLIPs on plasma pre-treated PE and PU substrates were produced using electrospun fiber mats consisting of a mixture PDMS and PA with the infusion of natural black seed oil (BSO). BSO was selected as a lubricant because of its natural character and antimicrobial properties containing 32 compounds, including 9-eicosyne (63.04%), linoleic acid (13.48%) and palmitic acid (9.68%) as major constituents [54,55,56]. Such production of SLIPs was unique to our knowledge, and this study provides a complex approach for the preparation of bacteria-free surfaces applicable in medical applications.

## 2. Materials and Methods

### 2.1. Material

Commercial grade of low-density polyethylene (LDPE) FE8000 pellets kindly provided by Qatar Petrochemical Company (QAPCO, Doha, Qatar) were used for a preparation of polymeric substrate for modification. Thin homogenous films of thickness approximately 0.41 mm were prepared by compression molding using an industrial mounting press machine (Carver, Wabash, IN, USA). The pellets were melted at 160 °C, compressed for 2 min using a force of 2 Tons, while maintaining the same temperature to obtain a flat surface. The LDPE films were cooled down to RT by water.

Polyether-based polyurethane (PU) films of approximately 0.21 mm thick provided by American Polyfilm (American Polyfilm Inc., Branford, CT, USA) were used as a polymeric substrate for modification.

The PE and PU films described above were washed by acetone and ethanol, respectively, to eliminate any dusts or possible contaminations from the production process that might affect the surface properties and were air-dried for 20 min at room temperature (RT). The dried samples were then cut out (10 cm × 10 cm) and directly used for surface treatment and various analyses.

Polydimethylsiloxane (PDMS) SYLGARD^®^ 184 elastomer kit purchased from Dow corning (Midland, MI, USA) described as Two-part, clear, 10:1, RT and heat cure, good strength, UL and Mil Spec was used for the production of fiber mats.

Copolyamide 6,10 (PA) (Vestamelt X1010, EVONIK Industries, Essen, Germany) was used for the production of electrospun fiber mats.

Isopropyl alcohol (IPA) purchased from BDH (Radnor, PA, USA), anhydrous (≥99% purity), inhibitor-free, was used as solvent for the preparation of PDMS/PA solutions.

Black seed oil (BSO) (AL Mashreq Int., 125 mL, Doha, Qatar) was used as a lubricant for ensuring slippery behaviour.

### 2.2. Procedure

#### 2.2.1. SLIPSs Preparation

The scheme of SLIPSs preparation on the PE and PU substrate is shown in Figure 1. The surface of PE a PU substrates was first plasma treated in order to improve adhesion properties between substrates and electrospun fiber mats. Electrospinning technique was subsequent use to fabricate porous structures (fiber mats) on the plasma pre-treated PE and PU substrates. Finally, natural BSO was infused into electrospun fiber mats to ensure slippery behavior.

#### 2.2.2. Plasma Treatment

A Venus75-HF plasma system (PlasmaEtch, Carson, CA, USA) was used for plasma treatment of PE and PU substrates under vacuum (~27 Pa) and radio-frequency (13.56 MHz). This equipment is consisted of a cylindrical chamber made of aluminum (25 cm in diameter and 28 cm deep). A capacitive parallel plate design generates plasma discharge with maximal nominal power of 120 W. All processing plasma parameters are controlled by the PC to ensure a high reproducibility of treatment process. First, the plasma parameters such as nominal power and treatment time were optimized in terms of achievement maximal wettability. The optimized plasma treatment time in air for PE and PU was 120 s (80 W nominal power) and 180 s (80 W nominal power), respectively [57,58].

#### 2.2.3. Solution Preparation and Electrospinning Procedure

To produce PDMS/PA fiber mats, a NaBond (Shenzhen, China) electrospinning device was used. Aluminum foil placed on drum was used as collector of fibers to ensure conductive substrate. Moreover, the PE and PU substrates were then fixed on the aluminum foil in order to fabricate porous structures on polymeric substrates. For the electrospinning process, a 5 mL syringe containing steel nozzle (22 g × 1 1/14,” 0.432 mm ID, 0.719 OD) was filled out with PDMS/PA/IPA solution and the electrospinning time was approximately 2 h in multi-jet mode. The PDMS (10 ppw) and curing agent (1 ppw) were mixed at 55 °C min for 30 min and as a mixture with PA, different concentrations were prepared using IPA as solvent (Table 1). The electrospinning process applied for the production of PDMS/PA fiber mats was carried out using 12.5–15 kV voltage; the distance needle/collector was 15 cm and flow rate was 2.5 mL/h.

#### 2.2.4. Oil Infusion

To create a slippery surface, natural BSO was infused into the electrospun fiber mats using the spin coating technique. The oil infusion into the electrospun fiber mats was carried out using Spin coating WS 650-23 (Spincoater- Laurell, North Wales, PA, USA) at speed of 500 rpm for 60 s. This system allows the production of uniform thin film layer by depositing a small drop of BSO onto the center of a substrate and then spinning the substrate at high speed. Moreover, this ensures the removal of oil excess from the substrate.

#### 2.2.5. Wettability and Sliding Behavior Analysis

The wettability of prepared electrospun fibers were evaluated by the water contact angle measurement. The optical contact angle measuring system OCA35 (DataPhysics, Filderstadt, Germany) was used for this study, which was equipped with a high resolution CCD camera. Ultra-pure water was used as testing liquid. A water droplet of approximately 1 µL was dispensed on the sample at ambient air conditions to eliminate gravitational effect. The contact angle was calculated approximately after 3 s (reaching of thermodynamic equilibrium between the liquid and the sample interfaces). Five independent measurements from different positions were carried out to obtain the average value of water contact angle. Moreover, the slippery effect of the electrospun fibers with infused oil was analyzed using the tilted stage at an angle of 10°. The water droplet was placed at 0° on the sample, and 10° was reached by slowly tilting of the whole equipment with the stage.

#### 2.2.6. Chemical Composition Characterization

The chemical composition of the fiber mats was evaluated by Fourier transform infrared spectroscopy with attenuated accessory (FTIR-ATR). The Spectrum 400 (Perkin Elmer, Waltham, MA, USA) was used to identify the functional groups and to characterize the chemical compositions of the prepared samples. All measurements were obtained through 8 scans with a resolution of 4 after background subtraction. Qualitative information about the absorbance of chemical groups in the middle infrared region (4400–550 cm^−1^) was obtained.

#### 2.2.7. Surface Morphology/Topography Characterization

The surface topographies of electro-spun fibers were analyzed using an optical surface metrology confocal system Leica DCM8 (Leica microsystems, Wetzlar, Germany). The optical system was used for high accuracy surface profiling to investigate the morphological changes induced by plasma treatment on the films surfaces. Images of size 160 µm × 130 µm were scanned using 100X objective lens. The surface roughness was quantitatively characterized in terms of the arithmetical mean height (Sa), which was calculated over the entire measured area.

The surface morphology after plasma treatment was studied using scanning electron microscopy (SEM). The SEM microscope Nova NanoSEM 450 (FEI, Hillsboro, OR, USA) was used to obtain 2D surface morphology of the analyzed surfaces. To get higher resolution images, thin Au layers a few nanometers thick were sputter-coated onto prepared fibers on aluminum, PE and PU samples and also to avoid the accumulation of electrons in the measured layer.

The detailed information about 3D surface topography of the fibers was obtained using atomic force microscopy (AFM). The AFM device MFP-3D (Oxford Instruments Asylum research, Abingdon, Oxford, UK) was employed in these experiments. Scanning was carried out under ambient conditions by a Silicon probe AC160TS (Al reflex coated Veeco model—OLTESPA, Olympus, Tokyo, Japan) in the tapping mode in air (AC mode) allowing obtaining images with from the surface area 20 µm × 20 µm. To obtain the proper topography of the fibers through AFM, a thin layer of fibers was attached to Scotch adhesive tape to ensure flat and smooth surface necessary for the measurement by this technique.

#### 2.2.8. Mechanical Properties Investigation

To identify changes in the mechanical properties of the prepared fibers, a nano-indentation technique was used. This technique allows measuring of indentation depths on the surface of thin sections by the application of very small forces to the indentation tip. The MFP-3D NanoIndenter (Asylum Research, USA) using a standard Berkovich indenter tip with a spring constant of 3940 N/m (calibration with Sapphire standard) was applied. Different areas (10 indentations) were examined with a loading force from 0 to 50 μN at a loading rate of 10 μN/s, an indentation force for 5 s; and unloading from 50 to 0 μN at a 10 μN/s unloading rate. The hardness (H) (Equation (1)) and the reduced modulus (Ec) (Equation (2)) of prepared fibers were then calculated using Oliver-Pharr method [59] considering an elastic behavior during the initial stages of unloading.
(1)$$H=\frac{P_{\max}}{A}$$
where P_max_ is the peak indentation load and A is the projected area of the hardness impression.
(2)$$\frac{1}{E_{c}}=\frac{\left(1-\nu^{2}\right)}{E}+\frac{\left(1-\nu_{i}{}^{2}\right)}{E_{i}}$$
where *E* and ν are Young’s modulus and Poisson’s ratio for the specimen and *E*_i_ and ν_i_ are the same parameters for the indenter tip.

#### 2.2.9. Adhesion Investigation

The study of adhesion characteristics at the macro level was done using a standard test method ASTM D6862. A peel resistance at 90° of peeling between PDMS/PA fiber mats and the PE or PU substrate was performed using a Peel Tester LF-Plus (Lloyd Instruments, West Sussex, UK) equipped with a 1 KN cell. The adhesion joints (19 mm width) were delaminated at 10 mm/min to ensure a steady peeling. Six independent measurements were performed to obtain average values for peel resistance.

AFM (MFP-3D (Asylum research, USA) was also used to analyze the adhesion characteristics of the prepared samples in the nano-scale using a force volume measurement technique. Adhesion force mapping was carried out in a contact mode using AC160TS cantilever (Al reflex coated Veeco model—OLTESPA, Olympus). Height and adhesion force mapping was done by the application of 32 × 32 contacts between AFM tip and the sample from the 20 µm x 20 µm surface area.

#### 2.2.10. Antimicrobial Tests

We modified a test method for an evaluation of the antimicrobial activity of modified plastic materials: ISO 22196 was used for an investigation of antimicrobial effects of the prepared samples. Samples were placed in sterile Petri dishes. Samples with dimensions 25 mm × 25 mm were then inoculated using standardized bacterial suspensions of *Staphylococcus aureus* (CCM 4516—2.3 × 10^5^ cfu/mL) or *Escherichia coli* (CCM 4517—7.1 × 10^6^ cfu/mL) 100 mL, and the samples were covered by ethanol disinfected polypropylene foil with dimensions 20 mm × 20 mm. The next step was an incubation of the inoculated samples at 100% of relative humidity and 35 °C for 24 h. The polypropylene foil was then removed, and samples were subsequently imprinted on agar (3 times on different areas) with lecithin 0.7 g/L and Tween 80 5 g/L as neutralizers, and incubated at 35 °C for 24 h. Then the number of bacterial colonies was evaluated based on scale from 0–5, wherein 0 represents the best antimicrobial effect without bacteria growing and vice versa.

## 3. Results

### 3.1. Fiber Mat Optimization

#### 3.1.1. Surface Morphology/Topography Study

Electrospun fiber mats composed from PDMS and PA were produced as hydrophobic porous structures used for subsequent oil infusion, resulting in SLIPSs. To overcome difficulties with electrospinning process of pure PDMS associated with its relatively low molecular weight, a mixture with PA has been used [60]. Different concentrations of the PDMS/PA/IPA solution were tried in order to optimize the fiber mats’ production processes (Figure 2). First, the two optimized concentrations of PA/IPA were tested before using PDM/PA/IPA mixtures. The concentration ratios of PA/IPA 1:10 and 1:20 (g/mL) solutions had good efficiency for the formation of smooth, homogeneous, bead-free fibers with relatively small fiber diameters (Table 2). However, PA mixed with PDMS using IPA in a ratio 1:1:10 (g/g/mL) led to the formation of regular but fewer fibers containing layers with larger fiber diameters. With more diluting of the mixture to PDMS/PA/IPA-1:1:20 (g/g/mL), the fibers were formed as thin, well-formed and bead-free in many layers. Another PDMS/PA/IPA ratio—1:0.5:20 (g/g/mL) with less PA content—led to the formation of quite regular fibers, but their quantity was relatively low. Figure 3 shows SEM-EDX mapping of electrospun fiber mats containing PDMS/PA/IPA-1:1:20 (g/g/mL) confirming presence of C, O, N and Si elements. Both polymers, PDMS and PA, contribute to C and O elements due to their chemical structures. Uniform, random distributions of N and Si, which represent PA and PDMS, respectively, within electrospun mats, are visible from Figure 3a, and indicate good miscibility of PDMS and PA polymers. Distinguishable, individual N and Si element maps in Figure 3b,c confirm the homogenous distribution of PA and PDMS polymers within the electrospun fiber mats. The optimized PDMS/PA/IPA-1:1:20 (g/g/mL) solution on the Al substrate was then applied for the fiber mats’ production on the plasma pre-treated PE and PU substrates.

The optimized electrospun fiber mats prepared using PDMS/PA/IPA 1:1:20 (g/g//mL) have been analyzed by profilometry and AFM in order to obtain information about their 3D surface topographies, and to study the surface topographies of the fibers’ structures for a larger surface area (876.6 µm × 659.8 µm) and the thickness using line profile (Figure 4a). The produced electrospun fiber mats exhibited regular bead-free and defect-free structures, and the overall layer thickness was about 3 µm.

The AFM analysis has been performed to provide detailed investigation of their 3D topography PAP alongside the line profile (Figure 4b). This has been useful for obtaining information about a fiber diameter and other fiber structural features, such as smoothness. The surface morphology/topography of fibers were measured in the area of 20 × 20 µm^2^ and the line profile graph was obtained from the Z-sensor (precise piezo response) images. The diameter of the fibers was taken as the average measurement of the height of individual peaks from the profile graph. The AFM proved regular structures of the resulting PDMS/PA fibers with regard to smoothness and distribution formation. The diameter range for electrospun fibers was 200–600 nm.

#### 3.1.2. Mechanical Properties Investigation

The mechanical properties are important to understand the fibers’ potential abilities. Therefore, these investigations were done thoroughly using the nanoindentation measurements by AFM with MFP-3D NanoIndenter. Hardness and reduced Young’s modulus (*E_c_*) were measured and calculated as penetration of indenter tip into the fibers’ surfaces in different areas and taking an average of 10 points. This allowed obtaining average values of hardness and *E_c_* with standard deviations, as shown in Table 3. The relation between stiffness and hardness can vary based on the properties in each material. For instance, PA fibers are assigned to have good mechanical properties and high resistance to abrasion. The reason behind it is the hydrogen bonding between amide group chains in PA [61]. In practice, the *E_c_* of the produced PA fibers prepared in IPA solvent was 4.08 MPa and their hardness was 0.40 MPa. In coexistence with PDMS, the *E_c_* and hardness of PDMS/PA decreased. The *E_c_* value was almost two and half times less than in the neat PA, achieving a value of 1.66 MPa, and hardness was reduced around three times, achieving a value of 0.15 MPa. Compared to the literature, the mechanical properties of the produced fibers were worse [62,63,64,65]. That can be related to the orientation of the PA chain during the electrospinning due to the solvent type, duration time and flow rate of the solution [58].

#### 3.1.3. Adhesion Characteristics

The plasma treatment was used in order to improve wettability and adhesion characteristics of PE and PU substrates prior to deposition of PDMS/PA fiber mats. Analyses of adhesion between polymeric substrates and produced electrospun fiber mats were carried out by peel test measurements using Scotch tape. Figure 5 shows peel resistance (peel force per entire width) of PA/PDMS fiber mats produced on the PE and PU substrates. Peel resistance of untreated PE was relative low achieving value 3.7 N/m. The plasma treatment was responsible for a slight improvement in the adhesion of PE, while peel resistance increased to 9.4 N/m. More visible differences were observed for PU. The peel resistance of untreated PU was much lower compared with untreated PE, achieving 0.03 N/m because of lower surface free energy of PU. Plasma treatment led to a remarkable increase in adhesion, while peel resistance increased to 2.2 N/m. The adhesion improvement of plasma treated polymeric surface was caused by the wettability improvement as result of functionalization and roughness changes [57,58].

### 3.2. Chemical Composition Characterization

FTIR has been used to analyze the chemical compositions of the PDMS/PA electrospun fiber mats. As PDMS was incorporated with PA, the presence of both constituents in the resultant fibers’ structures was proven. Figure 6 shows FTIR spectra of neat PA and PDMS/PA electrospun fiber mats. The spectrum of PA is characterized by specific absorbance bands [66]. The main absorbance bands observed in the neat PA were the –NH peak at 3300 cm^−1^, C=O at 1650 cm^−1^, symmetrical and asymmetrical –CH stretching at 2900 to 2980 cm^−1^ and N–H stretching at 1600 cm^−1^. As PDMS was mixed with PA in one solution and electro-spun fibers were produced, a spectrum of PDMS/PA showed the new absorbance bands and merged ones of both polymers, proving the existence of both PDMS and PA in the prepared electrospun fibers. The PDMS absorbance bands include Si–CH_3_ at 1270 and 825 cm^−1^, and O–Si–O at 1100 cm^−1^.

### 3.3. Characterization of SLIPs

#### 3.3.1. Slippery Behavior Investigation

Natural based BSO infused into the PDMS/PA electrospun fiber mats has been studied in terms of slippery behavior. The step oil infusion was carried out by drop casting followed by spin coating (detailed procedure is given in the experimental section). BSO was casted into electrospun fiber mats produced on plasma treated PE and PU substrates using PDMS/PA/IPA-1:1:20 (g/g/mL). Thus, modified samples exhibited satisfactory sliding behavior with water as the impinging liquid with contact angles of 56° and 65° on PE and PU substrates, respectively (Figure 7). A contact line pinning effect has been minimized using 3 µL of water (droplet) by minimizing gravity [67]. From the instrumentation and experimental point of view, the impinging water droplet was dispensed on the substrate at a 0° tilting, and the stage was inclined to 10° immediately after dispensing. The tilting from 0° to 10° proceeded for 50 s to avoid a vibration effect on the sliding behavior. It is apparent that the water droplet on the modified PE substrate slid faster than on the PU substrate. Intellectually, as the fibers produced on the substrates excelled in hydrophobic character with low wettability (contact angle of water ~ 135°), their behavior after oil infusion diverged, while the water exhibited a lower contact angle value. However, the oil infusion showed significant repellency to the impinged liquid by the factor of low adhesion between water droplet and oil surface. The low value of contact angle hysteresis Δθ = θa−θr, where θa represents advancing contact angle and θr is the receding contact angle (the actual measured Δθ = 3.0° to 4.5°), proved the ability of BSO to prevent the adhesion of water droplet on the substrate. This may refer to combination of the effect of surface tension of BSO and surface free energy of electrospun fiber mats acting as a repellence layer for water [33,68]. Observations revealed highly appreciable slippery behavior for PE and PU modified by PDMS/PA/BSO.

#### 3.3.2. Surface Morphology/Topography Investigation

SEM images at two various magnifications (1000× and 20,000×) of the PE and PU substrates modified by PDMS/PA/BSO are shown in Figure 8. BSO infusion caused covering of fiber mats structures, and the presence of the fiber structures was observed too. AFM was conducted to further detail the characterization of the prepared SLIPs from the 20 × 20 µm^2^ surface area in terms of the surface morphology/topography, and the results of amplitude images along with 3D height images and line profiles are shown in Figure 8. PDMS/PA/BSO produced on the PE substrate revealed a smoother surface in comparison to the same produced on the PU substrate. However, the presence of circle-shaped structures was observed. The irregularities of SLIPs on the PE substrate achieved a maximum about 600 nm in comparison to the 2000 nm of SLIPs produced on the PU substrate. This was probably main reason for differences in sliding behavior.

#### 3.3.3. Surface Adhesion Investigation

The adhesion phenomenon includes chemical and physiochemical factors. The adhesion properties of prepared SLIPSs were analyzed using AFM force mapping mode. Adhesion between the AFM tip and the sample surface at the nano-scale level (tip radius ~ 8 nm) was obtained using 32 × 32 independent contact interactions. The height/adhesion distribution images with line profiles are shown in Figure 9. The adhesion between AFM tip and electrospun fiber mats prepared on the PE (Figure 9a) and PU (Figure 9b) was relatively low, whereas adhesion between the AFM tip and fiber (approximately in the middle) had maxima of about 20 and 25 nN for PDMS/PA on PE and PDMS/PA on PU, respectively. The infusion of BSO into electrospun fiber mats led to decreases in adhesion characteristics. The maximum adhesion values between the AFM tip and fiber (middle area) were 12 nN and 2 nN for PDMS/PA/BSO on PE and PU, respectively. 

#### 3.3.4. Antibacterial Activity

The antimicrobial capabilities of SLIPs prepared on the PE and PU substrates have been tested against Gram-positive *S. aureus* and Gram-negative *E. coli* using intensive microbial activity assays, and the results are summarized in Table 4. The untreated PE and PU showed no resistance or inhibition to bacterial growth. This was due to the poor bacteria inhibition properties of PE and PU resulting from their chemical properties. Their subsequent modifications by PDMS/PA fiber mats did not improve antimicrobial properties. On the other hand, the PE and PU samples modified with PDMS/PA/BSO proved remarkable antimicrobial properties for both *S. aureus* and *E. coli*, as shown in Figure 10 and Figure 11, respectively. The best antimicrobial effects on both bacteria were observed for SLIPs prepared on the PE substrate, which also had the best sliding behavior.

## 4. Conclusions

This study has been focused on the preparation of SLIPs on the polymeric substrates and their effects on the antimicrobial activity. It has been found that the optimal concentration of PDMS/PA/IPA-solution for the production of electrospun fiber mats (porous structures) using electrospinning technique was 1:1:20 (g/g/mL), which was applied for deposition on the plasma treated PE and PU substrate. Plasma treatment acted as adhesion promoter between polymeric substrate and PDMS/PA electrospun fiber mats, which was confirmed by peel resistance measurements. The subsequent infusion of natural BSO into the porous structures caused slippery behavior, as was confirmed by sliding angle measurements at 10° of titling angle. Thus, modified PE and PU surfaces proved significant antimicrobial activity against Gram-positive *S. aureus* and Gram-negative *E. coli* bacteria.

## Figures and Tables

**Figure 1 polymers-13-00206-f001:**
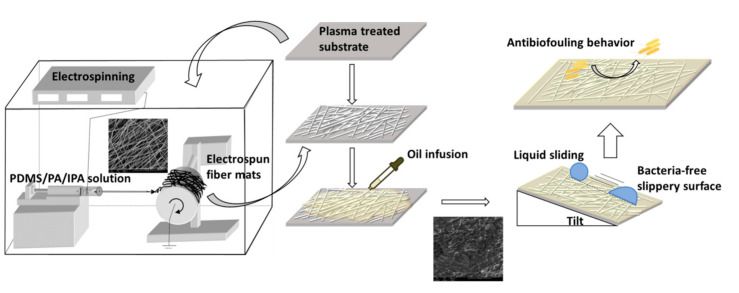
Electrospinning and oil infusion procedures on PE and PU substrates.

**Figure 2 polymers-13-00206-f002:**
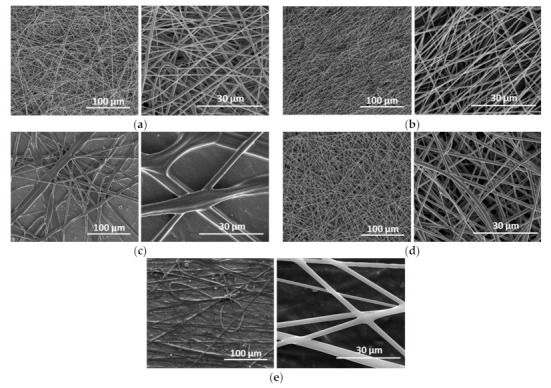
SEM images of the PDMS/PA fiber mats produced on Al: (**a**) PA/IPA—1:10 (g/mL), (**b**) PA/IPA—1:20 (g/mL), (**c**) PDMS/PA/IPA—1:1:10 (g/g/mL), (**d**) PDMS/PA/IPA—1:1:20 (g/g/mL), (**e**) PDMS/PA/IPA—1:0.5:20 (g/g/mL).

**Figure 3 polymers-13-00206-f003:**
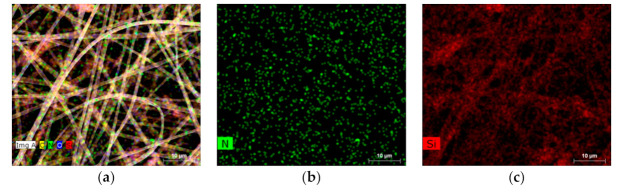
SEM-EDX images with mapping of chemical elements of the optimized PDMS/PA/IPA-1:1:20 (g/g/mL) fiber mats: (**a**) C, O, N, Si, (**b**) N, (**c**) Si.

**Figure 4 polymers-13-00206-f004:**
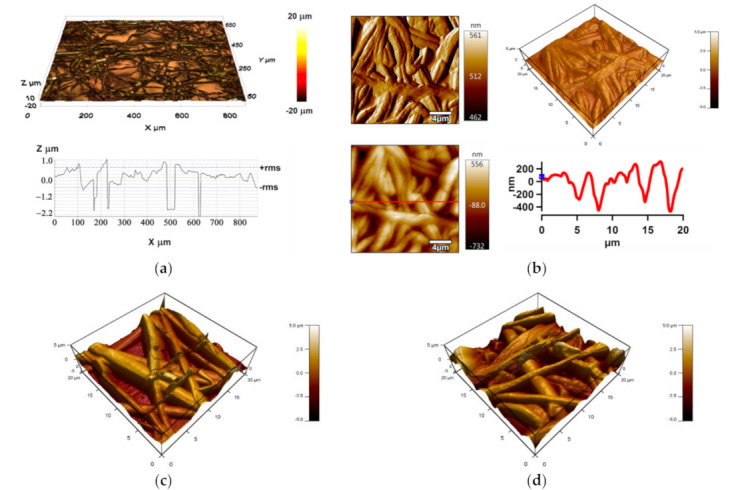
Profilometry image (**a**), AFM images—amplitude and 3D height, and a Z-sensor with line profile of PDMS/PA electrospun fiber mats prepared on aluminum substrate (**b**); and 3D height AFM images of electrospun fiber mats prepared on PE (**c**) and PU (**d**) substrates.

**Figure 5 polymers-13-00206-f005:**
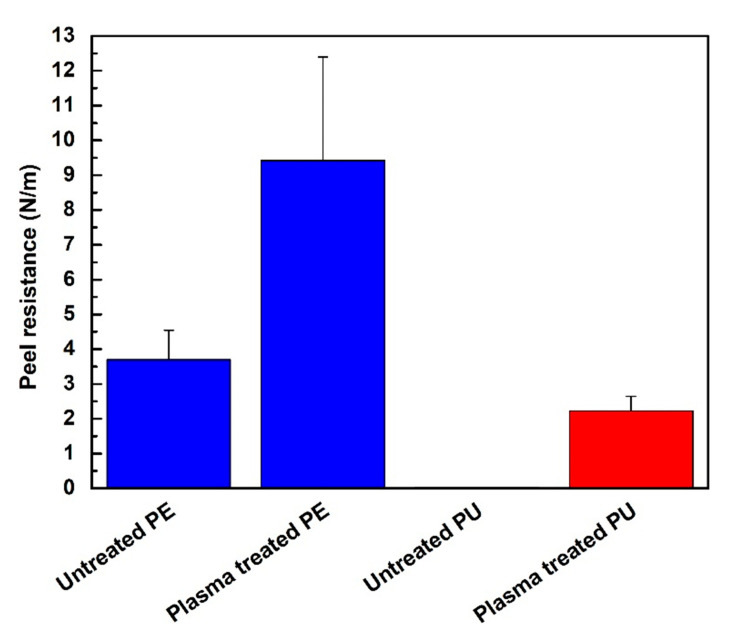
Peel resistance for PDMS/PA electrospun fiber mats of PE and PU.

**Figure 6 polymers-13-00206-f006:**
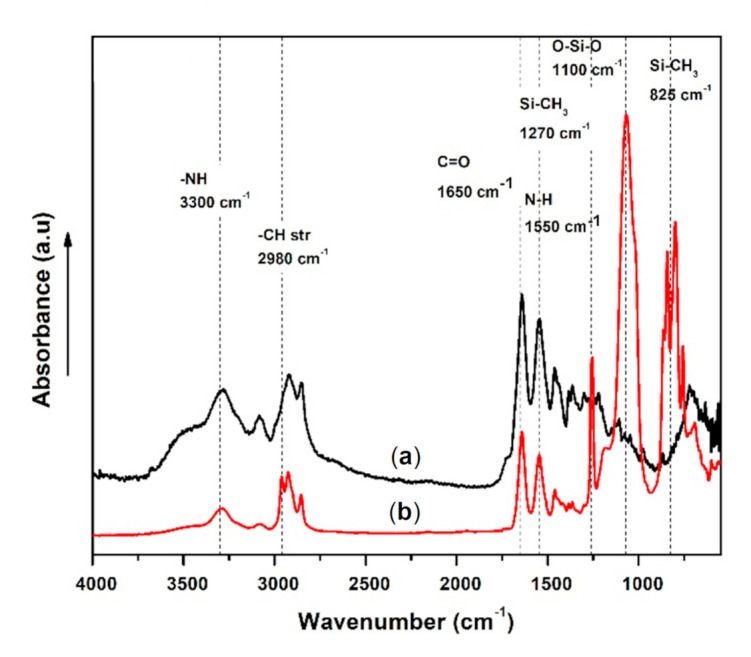
FTIR spectra of PA (**a**) and PDMS/PA (**b**) electrospun fiber mats.

**Figure 7 polymers-13-00206-f007:**
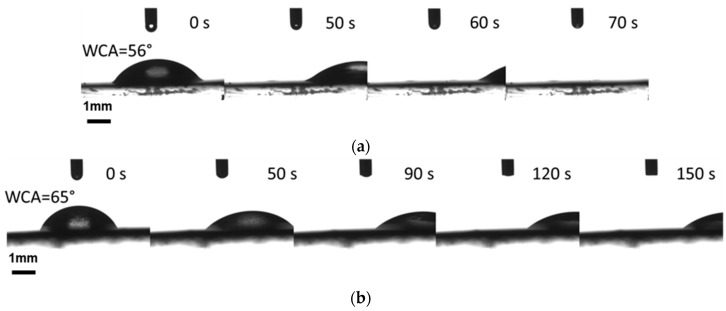
Sliding of water droplet at 10° tilting on PDMS/PA/BSO-PE (**a**) and PDMS/PA/BSO-PU (**b**).

**Figure 8 polymers-13-00206-f008:**
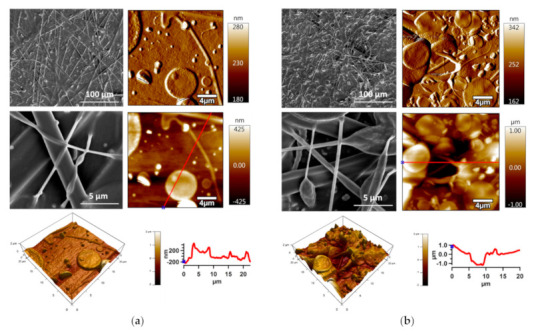
SEM and AFM images of: (**a**) PDMS/PA/BSO-PE, (**b**) PDMS/PA/BSO-PU.

**Figure 9 polymers-13-00206-f009:**
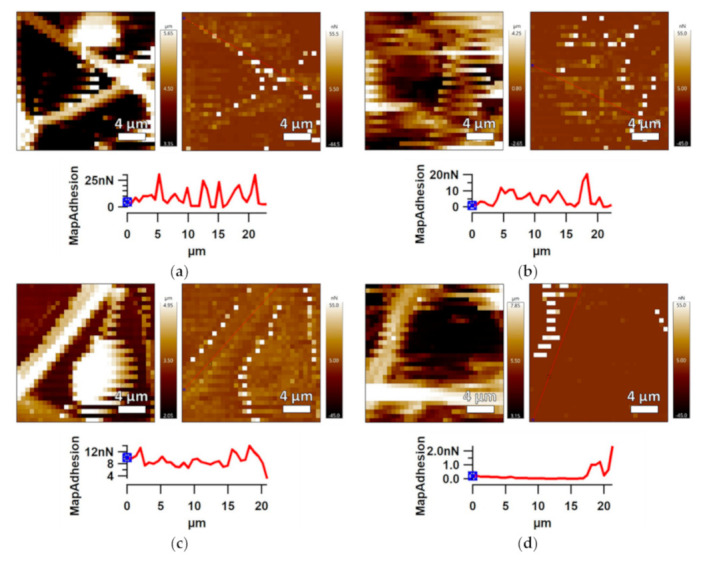
AFM force mapping images (height—left, adhesion—right) of: (**a**) PDMS/PA-PE, (**b**) PDMS/PA-PU, (**c**) PDMS/PA/BSO-PE, (**d**) PDMS/PA/BSO-PU.

**Figure 10 polymers-13-00206-f010:**
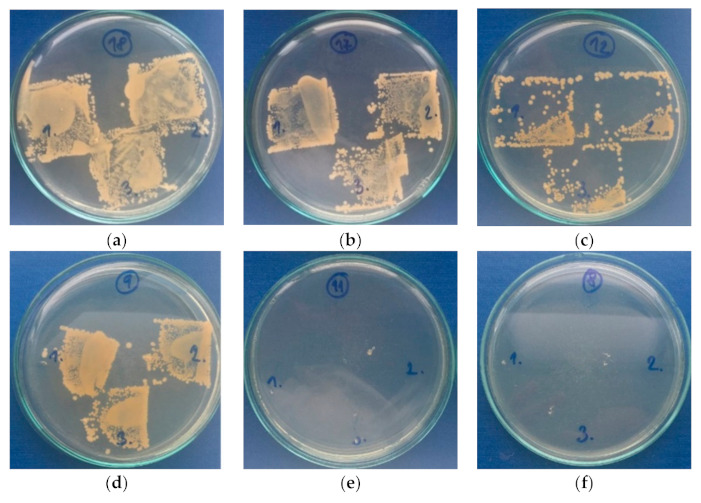
Total microbial counts on plate-count agar with inoculated *Staphylococcus Aureus*: (**a**) PE, (**b**) PU, (**c**) PDMS/PA-PE, (**d**) PDMS/PA-PU, (**e**) PDMS/PA/BSO-PE, (**f**) PDMS/PA/BSO-PU.

**Figure 11 polymers-13-00206-f011:**
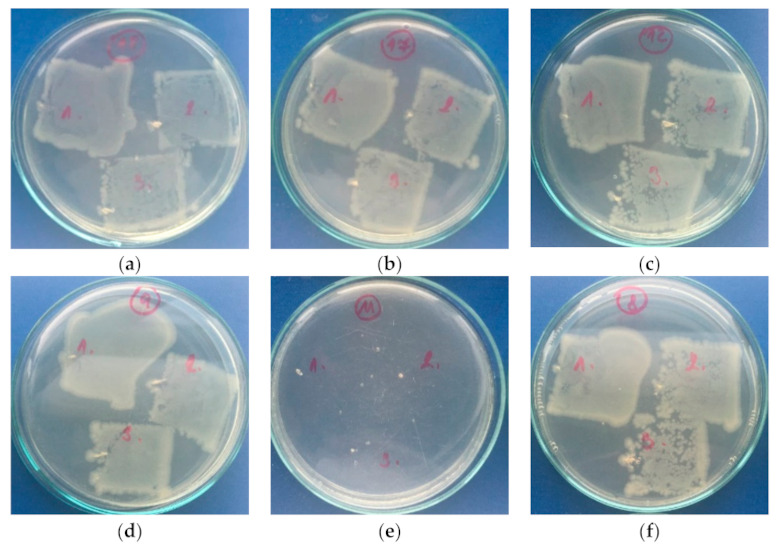
Total microbial counts on plate-count agar with inoculated *Escherichia Coli*: (**a**) PE, (**b**) PU, (**c**) PDMS/PA-PE, (**d**) PDMS/PA-PU, (**e**) PDMS/PA/BSO-PE, (**f**) PDMS/PA/BSO-PU.

**Table 1 polymers-13-00206-t001:** Parameters/conditions for preparation/electrospinning of PA/PDMS.

**Polymer**		**Solvent**	**Solution**
PA (g)		IPA (mL)	Percentage (% *w/v*)
1		10	9
1		20	5
**Polymer Mixture**		**Solvent**	**Solution**
PA (g)	PDMS (g)	IPA (mL)	Percentage (% *w/v*)
1	1	10	17
1	1	20	9
0.5	1	20	7

**Table 2 polymers-13-00206-t002:** Thickness of electrospun fibers.

Sample	Fiber Diameter (µm)	SD (µm)
PA/IPA—1:10 (g/mL)	0.730	0.156
PA/IPA—1:20 (g/mL)	0.533	0.144
PDMS/PA/IPA—1:1:10 (g/g/mL)	3.021	0.988
PDMS/PA/IPA—1:1:20 (g/g/mL)	1.174	0.300
PDMS/PA/IPA—1:0.5:10 (g/g/mL)	2.381	0.908

**Table 3 polymers-13-00206-t003:** The reduced Young’s modulus E_C_ and the hardness of the electrospun fibers.

Sample	E_C_ (MPa)	SD	Hardness (MPa)	SD
PA	4.08	1.40	0.40	0.11
PDMS/PA	1.66	0.37	0.15	0.03

**Table 4 polymers-13-00206-t004:** Antimicrobial activity of prepared samples.

Sample	Bacterial Colonies Increase *	
	*S. aureus* CCM 4516	*E. coli* CCM 4517
PE	4–5, 5, 4–5	4–5, 4–5, 5
PU	4, 4–5, 4–5	4–5, 4–5, 4–5
PA/PDMS-PE (no oil)	2, 2–3, 3	4–5, 4–5, 4–5
PA/PDMS-PU (no oil)	4–5, 4–5, 4–5	5, 5, 5
PA/PDMS/BSO-PE	0, 0, 0	0–1, 0, 0
PA/PDMS/BSO-PU	0, 0, 0	4, 0–1, 4

* The scale for assessing the growth of bacterial colonies: 0—without growth, 1—detectable amount (single colony), 2—detectable amount (combined colony), 3—second imprint—distinguishable colonies, third imprint can be detected, 4—third imprint—distinguishable colonies, 5—overgrown—continuous growth.

## Data Availability

MDPI Research Data Policies.

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
