# Peer review of "Slippery Liquid-Infused Porous Polymeric Surfaces Based on Natural Oil with Antimicrobial Effect"

_polymers, 2021, doi:10.3390/polym13020206_

Round 1

Reviewer 1 Report

Title: Slippery liquid-infused porous polymeric surfaces based on natural oil with an antimicrobial effect

Authors: Salma Habib, Sifani Zavahir, Aya E. Abusrafa, Asma Abdulkareem, Patrik Sobolčiak, Marian Lehocky, Daniela Vesela, Petr Humpolíček and Anton Popelka

The manuscript presents the fabrication and characterization of polymer based slippery liquid-infused porous surfaces (aka SLIPS). Additionally, the authors characterized the anti-fouling behavior of the surface via antimicrobial adhesion using Gram-positive S. aureus and Gram-negative E. coli bacteria.

The focus of the paper is not obvious. Neither is it stated clearly. The authors spend the majority of the time discussing the fabrication steps and characterizations that are not relevant to the story. Surprisingly, the supposedly focus of the work (antimicrobial adhesion) is discussed in one short paragraph on page 12 section 3.4.2. More importantly, despite the major claim (reduced antimicrobial adhesion), adhesion data (adhesion force measurement) is not provided.

The manuscript is not well written. It lacks focus and the story is not clear. Its technical content is poor with insufficient discussion. Moreover, the manuscript suffers from ill sentence construction and numerous grammatical errors. The introduction section is inadequate, poorly written and missing important papers in the field. Overall, the work is not complete and requires a major overhaul. In its current form, it does not contribute to the knowledge base in the field.

For the reasons I pointed out in my assessment below, I do not recommend the manuscript for publication on Polymers.

Major comments:

  1. Page 5, line 196-198, “The contact angle was calculated after approximately 3 s (reaching thermodynamic equilibrium between the liquid and the sample interfaces).” Explain what thermodynamic equilibrium in this context means.
  2. Page 5, line 200-201, the authors stated a 10° tilting angle. This suggests that the water droplet is in contact with the underlying substrate by displacing the lubrication film. Is there any contact line pinning? What is the spreading coefficient of water on oil in this material design? Explain.
  3. Explain the rational for selecting Black seed oil? What is the chemical formula and composition of Black seed oil?
  4. Provide AFM measurements before and after oil impregnation. Figure 4 and Figure 8 do not compare well.

Minor comments:

  1. Explain the sentence on page 11 line 352-353 that reads, “This may refer to a combination of the effect of surface tension of BSO and surface free energy of electrospun fiber mats acting as a repellent layer for water.”
  2. Explain how the advancing and receding contact angles are measured. Provide data.
  3. On figure 7, how is contact angle measured on an inclined plane? Are the angles shown in the figure static/equilibrium contact angles? Explain.

General comments:

  1. Page 5, line 178, the statement that reads in part, “optimal ratio 10:1 at 55 °C min for 30 min …” is not clear.
  2. The font for axis label in Figure 4 is small and difficult to read.
  3. Page 2 line 46, the statement that reads, “Pathogenic biofilm is resists destruction …” is grammatically incorrect. There are numerous grammatically incorrect sentences.
  4. Page 7, line 253: milliliter is denoted as ml. However, on the same page line 258, it is denoted as mL. Use consistent symbols.
  5. Add a scale bar in Figure 7.
  6. Increase the font for Figure 8.

Author Response

The manuscript presents the fabrication and characterization of polymer based slippery liquid-infused porous surfaces (aka SLIPS). Additionally, the authors characterized the anti-fouling behavior of the surface via antimicrobial adhesion using Gram-positive S. aureus and Gram-negative E. coli bacteria.

The focus of the paper is not obvious. Neither is it stated clearly. The authors spend the majority of the time discussing the fabrication steps and characterizations that are not relevant to the story. Surprisingly, the supposedly focus of the work (antimicrobial adhesion) is discussed in one short paragraph on page 12 section 3.4.2. More importantly, despite the major claim (reduced antimicrobial adhesion), adhesion data (adhesion force measurement) is not provided.

Reply: The authors thank reviewer for all valuable comments. Reduced adhesion has been indirectly confirmed by sliding behavior of modified PE and PU substrates by SLIPS. The measurement of AFM tip interactions have been investigated and AFM force mapping images have been included in revised manuscript.

The manuscript is not well written. It lacks focus and the story is not clear. Its technical content is poor with insufficient discussion. Moreover, the manuscript suffers from ill sentence construction and numerous grammatical errors. The introduction section is inadequate, poorly written and missing important papers in the field. Overall, the work is not complete and requires a major overhaul. In its current form, it does not contribute to the knowledge base in the field.

Reply: The English grammar has been checked by MDPI during reviewing process and it is corrected in revised manuscript. Introduction has been extended by relevant references. 

For the reasons I pointed out in my assessment below, I do not recommend the manuscript for publication on Polymers.

Major comments:

  1. Page 5, line 196-198, “The contact angle was calculated after approximately 3 s (reaching thermodynamic equilibrium between the liquid and the sample interfaces).” Explain what thermodynamic equilibrium in this context means.

Reply: It means that equilibrium between soli-liquid, solid-vapor and liquid-vapor interactions have been stabile after 3 s, generally said the contact angle changed only slowly over time after 3 s. For the minimizing errors, repeatability and comparability of all experiments, contact angle has been measured at 3 s.

  1. Page 5, line 200-201, the authors stated a 10° tilting angle. This suggests that the water droplet is in contact with the underlying substrate by displacing the lubrication film. Is there any contact line pinning? What is the spreading coefficient of water on oil in this material design? Explain.

Reply: The contact line pinning effect has been minimized using 3  µl of water droplet to eliminate gravity forces. However some pinning effect has been observed, especially in PA/PDMS/BSO on PE substrate, which slightly affected water contact angle. The contact angle of water on neat oil at RT was 54.8±1.9°.

  1. Explain the rational for selecting Black seed oil? What is the chemical formula and composition of Black seed oil?

Reply: Black seed oil was selected out of 15 tested different natural oils in terms of good sliding behavior. Moreover, black seed oil excels by antimicrobial properties. The chemical composition of Black seed oil (below Table) according to S, Dinakaran et al. (DOI: 10.13040/IJPSR.0975-8232.7(11).4473-79):

S.no.

Retention Time

Chemical Name

Area %

1

2.016

2-Methylpentane

0.38

2

2.087

3-Methylpentane

0.39

3

2.414

Methylcyclopentane

0.88

4

2.761

Cyclohexane

0.52

5

9.277

4-methyl-1-(1-methylethyl),didehydro-biocyclo[3.1.0]hexane

0.67

6

9.440

(+)-α-Pinene

0.62

7

10.349

Sabinene

0.38

8

10.441

(-)-β-Pinene

0.37

9

11.442

P-Cymene

2.54

10

11.503

(-)-Limonene

0.35

11

12.055

γ-Terpinene

0.39

12

12.719

β-Terpinene

0.26

13

13.127

4,6-Diamino-5-formamidopyrimidine

0.33

14

13.883

Phellandral

0.13

15

14.223

Terpinen-4-ol

0.24

16

15.292

Thymoquinone

1.86

17

16.171

P-Thymol

0.07

18

16.742

(+)-α-Longipinene

0.06

19

17.702

Longifolene

0.24

20

20.419

Paeonol

0.11

21

23.248

(R)-(+)-β-Citronellol

0.05

22

23.850

Myristic acid

0.10

23

27.292

3-Todomethyl-3,6,6-trimethyl-cyclohexene

0.11

24

27.445

Cyclododecene

0.19

25

27.905

Palmitic acid

9.68

26

29.774

Methyl linoleate

0.14

27

29.917

3,5-Dimethylcyclohexanol

0.91

28

31.459

9-Eicosyne

63.04

29

31.643

Linoleic acid

13.48

30

34.226

Cs-7-Dodecen-1-yl acetate

1.11

31

35.493

Octadeca-9,17-dienal

0.11

32

36.146

Linoleic acid ethyl ester

0.29

  1. Provide AFM measurements before and after oil impregnation. Figure 4 and Figure 8 do not compare well.

Reply: Figure 4 has been updated with microscopic images of PA/PDMS prepared on PE and PU substrates.

Minor comments:

  1. Explain the sentence on page 11 line 352-353 that reads, “This may refer to a combination of the effect of surface tension of BSO and surface free energy of electrospun fiber mats acting as a repellent layer for water.”

Reply: This sentence expresses the combination of low surface tension of BSO as a direct contact layer with water and low surface free energy of electrospun fiber mats, which affects water adhesion. The sentence has been modified in revised manuscript for better clarity: “This may refer to a combination of the effect of low surface tension of BSO and low surface free energy of electrospun fiber mats acting as a repellent layer for water (low adhesion)”

  1. Explain how the advancing and receding contact angles are measured. Provide data.

Reply: Advancing and receding contact angles were measured during sliding angle measurements representing left (advancing) and right (receding) contact angle of water droplet placed on modified PE and PU substrates. The difference between advancing and receding contact angle was in the range 3.0˚ to 4.5˚, each value of contact angle of water was evaluated as an average of advancing and receding contact angle. For example, advancing and receding water contact angle was 53° and 59°, respectively, for PDMS/PA/BSO on PE substrate.

  1. On figure 7, how is contact angle measured on an inclined plane? Are the angles shown in the figure static/equilibrium contact angles? Explain.

Reply: Contact angle on inclined plane has been realized by video capturing over time and presented individual values of contact angles have been evaluated at 0 s prior to start tilting.

General comments:

  1. Page 5, line 178, the statement that reads in part, “optimal ratio 10:1 at 55 °C min for 30 min …” is not clear.

Reply: This statement has been restyled in revised manuscript “The PDMS (10 ppw) and curing agent (1 ppw) were mixed at 55 °C min for 30 min and as a mixture with PA, different concentrations were prepared using IPA as solvent (Table 1)”.

  1. The font for axis label in Figure 4 is small and difficult to read.

Reply: The font for axis label has been increased in Figure 4.

  1. Page 2 line 46, the statement that reads, “Pathogenic biofilm is resists destruction …” is grammatically incorrect. There are numerous grammatically incorrect sentences.

Reply: The English grammar has been checked by MDPI during reviewing process and it is corrected in revised manuscript. For example: “Pathogenic type of biofilm is resistive against destruction by the immune system/antibiotic leading to serious infections”

  1. Page 7, line 253: milliliter is denoted as ml. However, on the same page line 258, it is denoted as mL. Use consistent symbols.

Reply: The consistent symbols have been used in revised manuscript.

  1. Add a scale bar in Figure 7.

Reply: Scale bars have been included in Figure 7.

  1. Increase the font for Figure 8.

Reply: Fonts have been increased in revised manuscript.

Reviewer 2 Report

In this research, author made porous surface using electrospinning and impregnated natural oil into the surface to obtain slippery surface. The research is interesting. However, several concerns should be addressed for publication in polymers.

  1. To produce slippery surface, silicone oil was commonly used and the oil is infused into superhydrophobic surface to obtain stable and durable lubricant layer. In this study, a different approach was used to make slippery surface. Author needs to explain why natural oil was used and what selling point of using the oil is. Stability of slippery layer is important. Is it stable against external force or is there any leaching when it is dipped into water?

  1. At line 46. “Pathogens biofilm is resists” is grammatically incorrect. Please correct the sentence.  
  2. At line 144, what is cut sample size.
  3. Please add more detail of electrospinning. For example, nozzle size, spray mode (conjet ?? spindle??, multi-jet mode??)
  4. At line 213,225, 276, please correct a writing of dimensions it should be “um X um”, not “um2
  5. Figure 2 shows SEM image of surface produce by electrospinning and each figure show different size of fibre thickness. Author needs to provide average thickness with standard deviation and porosity information at each condition.
  6. In Table 2, PDMS addition into PA resulted in a significant reduction of Young’s modulus and hardness. Is it still usable??
  7. In wettability test, water contact angle of slippery surfaces was < 90 degree. This means that the surface is hydrophilic. But at line 346, author mentioned that surface shows excellent hydrophobic character. Author needs to select the term carefully.
  8. In terms of antimicrobial test, slippery surface normally has anti-biofouling effect. The term” antimicrobial” contains two meanings (bactericidal and anti-biofouling). Author did not explain experimental method in manuscript. It is confusing.  Author needs to clarify if it is bactericidal or anti-biofouling test.

Author Response

In this research, author made porous surface using electrospinning and impregnated natural oil into the surface to obtain slippery surface. The research is interesting. However, several concerns should be addressed for publication in polymers.

  1. To produce slippery surface, silicone oil was commonly used and the oil is infused into superhydrophobic surface to obtain stable and durable lubricant layer. In this study, a different approach was used to make slippery surface. Author needs to explain why natural oil was used and what selling point of using the oil is. Stability of slippery layer is important. Is it stable against external force or is there any leaching when it is dipped into water?

Reply: The black seed oil was selected because of its natural character (eco-friendly) and antimicrobial properties. The following statement has been included in revised manuscript: “BSO was selected as a lubricant because of its natural character and antimicrobial properties containing 32 compounds including 9-eicosyne (63.04%), linoleic acid (13.48%), palmitic acid (9.68%) as major constituents”. As blackseed oil has higher affinity to hydrophobic electrospun fiber mats, leaching has been not observed under water.

  1. At line 46. “Pathogens biofilm is resists” is grammatically incorrect. Please correct the sentence.  

Reply: The English grammar has been checked by MDPI during reviewing process and it is corrected in revised manuscript. For example: “Pathogenic type of biofilm is resistive against destruction by the immune system/antibiotic leading to serious infections”

  1. At line 144, what is cut sample size.

Reply: Samples were cut into 10 cm x 10 cm sheets. This information has been included in revised manuscript: “The dried samples were then cut out (10 cm x10 cm) and directly used for surface treatment and various analyses”.

  1. Please add more detail of electrospinning. For example, nozzle size, spray mode (conjet ?? spindle??, multi-jet mode??)

Reply: The detailed electrospinning information have been included in revised manuscript: ” For the electrospinning process, 5ml syringe containing steel nozzle (22g x 1 1/14”, 0.432 mm ID, 0.719 OD) was filled out with PDMS/PA/IPA solution and the electrospinning time was approximately 2 hours in multi-jet mode.

  1. At line 213,225, 276, please correct a writing of dimensions it should be “um X um”, not “um2

Reply: The dimensions writing format has been changed in revised manuscript.

  1. Figure 2 shows SEM image of surface produce by electrospinning and each figure show different size of fibre thickness. Author needs to provide average thickness with standard deviation and porosity information at each condition.

Reply: The table regarding electrospun fiber thickness with SD has been included in the revised manuscript.

  1. In Table 2, PDMS addition into PA resulted in a significant reduction of Young’s modulus and hardness. Is it still usable??

Reply: The reduction of Young’s modulus and hardness has been expected as PDMS is material with much worse mechanical properties in compare with PA. Nevertheless, Young’s modulus and hardness are still acceptable for these applications, while the mechanical properties are mainly carried by PE and PU substrates.

  1. In wettability test, water contact angle of slippery surfaces was < 90 degree. This means that the surface is hydrophilic. But at line 346, author mentioned that surface shows excellent hydrophobic character. Author needs to select the term carefully.

Reply: The statement about hydrophobic character belongs to only electrospun fiber mats without oil infusion achieving water contact angle around 135°. Black seed oil infusion into fiber mats resulted in decrease in contact angle becaues of low surface tension of oil.

  1. In terms of antimicrobial test, slippery surface normally has anti-biofouling effect. The term” antimicrobial” contains two meanings (bactericidal and anti-biofouling). Author did not explain experimental method in manuscript. It is confusing.  Author needs to clarify if it is bactericidal or anti-biofouling test.

Reply: The authors thank reviewer for noticing this missing information it has been included in revised manuscript: “Modified a test method for an evaluation of the antimicrobial activity of modified plastic materials ISO 22196 was used for an investigation of antimicrobial effect of prepared samples. Samples were placed in sterile Petri dishes. Samples with dimensions 25 mm × 25 mm were then inoculated using  standardized bacteria suspension of  S. aureus (CCM 4516 – 2.3×105 cfu/ml) and E. coli (CCM 4517 – 7.1×106 cfu/mL) 100 mL, and the samples were covered by ethanol disinfected polypropylene foil with dimensions 20 mm × 20 mm. The next step was an incubation of the inoculated samples at 100% of relative humidity and 35 °C for 24 hours. The polypropylene foil was then removed, and samples were subsequently imprinted on agar (3 times on different areas) including lecithin 0.7 g/L a Tween 80 5 g/L as neutralizations and incubated at 35 °C for 24 hours. Then the number of bacterial colonies was evaluated based on scaling from 0-5, while 0 represents the best antimicrobial effect without bacteria growing and vice versa”.

Round 2

Reviewer 2 Report

The concerns that I raised are addressed.   I think that the paper has enough novelty to publish in your journal.